# The AUX1-AFB1-CNGC14 module establishes a longitudinal root surface pH profile

**Nelson BC Serre[1†], Daša Wernerová[1,2†], Pruthvi Vittal[1], Shiv Mani Dubey[1], Eva Medvecká[1], Adriana Jelínková[3], Jan Petrášek[1,3], Guido Grossmann[2,4], Matyáš Fendrych[1]\***

[1]Department of Experimental Plant Biology, Faculty of Science, Charles University, Prague, Czech Republic; [2]Institute of Cell and Interaction Biology, Heinrich-Heine-University Düsseldorf, Düsseldorf, Germany; [3]Institute of Experimental Botany, Czech Academy of Sciences, Prague, Czech Republic; [4]CEPLAS - Cluster of Excellence on Plant Sciences, Heinrich-Heine-University Düsseldorf, Düsseldorf, Germany

**Abstract** Plant roots navigate in the soil environment following the gravity vector. Cell divisions in the meristem and rapid cell growth in the elongation zone propel the root tips through the soil. Actively elongating cells acidify their apoplast to enable cell wall extension by the activity of plasma membrane AHA H$^+$-ATPases. The phytohormone auxin, central regulator of gravitropic response and root development, inhibits root cell growth, likely by rising the pH of the apoplast. However, the role of auxin in the regulation of the apoplastic pH gradient along the root tip is unclear. Here, we show, by using an improved method for visualization and quantification of root surface pH, that the *Arabidopsis thaliana* root surface pH shows distinct acidic and alkaline zones, which are not primarily determined by the activity of AHA H$^+$-ATPases. Instead, the distinct domain of alkaline pH in the root transition zone is controlled by a rapid auxin response module, consisting of the AUX1 auxin influx carrier, the AFB1 auxin co-receptor, and the CNCG14 calcium channel. We demonstrate that the rapid auxin response pathway is required for an efficient navigation of the root tip.

## Editor's evaluation

All the results present solid evidence supporting the impact statement that 'Plant roots can rapidly change the acidity of their cell walls and the root-soil interface to efficiently navigate in the growing environment." These findings are important and have practical implications beyond Arabidopsis biology with potential future impacts in crop improvement, soil sciences and general plant physiology. The evidence is convincing and appropriately validated in line with current state-of-the-art.

**\*For correspondence:**
matyas.fendrych@natur.cuni.cz

†These authors contributed equally to this work

**Competing interest:** The authors declare that no competing interests exist.

## Introduction

Plants colonize soil by the growth of their roots. Because plant cells are non-motile, mechanically coupled by their cell walls and symplastically connected by plasmodesmata, the extent and direction of root growth is driven by the elongation of cells that can differ across the organ (*Braidwood et al., 2014*). The balance between cell proliferation and differentiation in the root apex is controlled by several phytohormone and peptide signaling pathways, among which auxin plays a specific role, thanks to its long-distance intercellular transport that serves as the information carrier between distant plant tissues (*Grieneisen et al., 2007*; *Wisniewska et al., 2006*).

The root apex is divided in several zones fulfilling various roles in root development. At the tip of the root, the root cap covers and protects the meristem, and is the center of gravity perception. In the meristem, stem cells produce transit-amplifying cells that populate the root tip (*Dolan et al., 1993*; *Campilho et al., 2006*). In the model plant *Arabidopsis thaliana*, the root cap reaches up to the transition zone, where cells stop dividing and prepare for elongation; the transition zone is characterized by a distinct physiology and shows a high responsiveness to external factors (*Verbelen et al., 2006*). Further in the shootward direction, cells rapidly elongate in the elongation zone, propelling the root tip through the soil (*Beemster and Baskin, 1998*). After reaching their final length, cells differentiate in the maturation zone, where the root hairs emerge presumably to increase the absorptive root surface.

To elongate, cells must expand their cell walls, increase turgor pressure or both (*Lintilhac, 2014*). Proton extrusion into the apoplast by the plasma membrane (PM) H⁺ ATPases is crucial for both cell wall expansion and turgor maintenance. Acidic pH leads to activation of cell wall remodeling enzymes, and the H⁺ gradient at the PM drives the transport of most nutrients and ions, and as such is required for turgor pressure generation (*Falhof et al., 2016*). It is therefore not surprising that root zones of numerous species show a distinct pattern of surface proton fluxes and cell wall pH (*Siao et al., 2020*). In general, the tip of the root and the late elongation and maturation zones show outward H⁺ fluxes, while the transition zone and early elongation zones show inward H⁺ fluxes (*Weisenseel and Meyer, 1997*). Several studies showed that this battery-like system, with positive charges flowing out of the maturation zone and entering at the transition zone, is mainly driven by H⁺ fluxes without completely excluding calcium and chloride ions (*Weisenseel et al., 1979*; *Björkman and Leopold, 1987*; *Behrens et al., 1982*; *Iwabuchi et al., 1989*). Similar pattern of H⁺ fluxes was determined in *A. thaliana*, where the fluxes also correlated with root surface pH: an alkaline domain was observed at the border between meristematic and transition zones (*Staal et al., 2011*). Other works have attempted to determine the pH of the epidermal cell walls, and show a general decrease of pH toward the maturation zone of the root (*Barbez et al., 2017*; *Moreau et al., 2022*; *Großeholz et al., 2022*). It is important to keep in mind that changes in proton fluxes and membrane potential are not always correlated with change in pH. For example, a simultaneous leak of protons and uptake through H⁺/K⁺ symporter in the cytoplasm can lead to apoplast acidification with a relatively stable membrane potential (*Stanković, 2006*). In general, the relationship between H⁺ fluxes, cell wall pH, and the growth rate in the particular zones remains unclear: the highest pH is observed in the transition and early elongation zones, while the lowest pH and highest outward H⁺ fluxes are observed in the maturation zone where cells do not elongate. In particular, the mechanism of the alkaline domain formation and its physiological significance remain unclear.

The apoplastic pH is largely controlled by the activity of the proton pump H⁺ ATPases, encoded by 11 genes in the *A. thaliana* genome; in roots, the dominant paralogs are AHA1 and AHA2 (*Haruta et al., 2010*). While their activity is mainly regulated by phosphorylation (*Falhof et al., 2016*), the expression and membrane localization of AHAs along the longitudinal root axis seems to be important for the zonation of the *A. thaliana* root. *Haruta et al., 2018*, showed that under conditions of dim light, the AHA2-mCitrine was less abundant in the membranes of transition zone cells, and this partially correlated with higher surface pH of the transition zone cells. Under conditions of light, the AHA2-mCitrine localizes to the membranes in the transition zone. *Pacifici et al., 2018*, presented a model where cytokinin signaling regulates root meristem size by controlling the expression of AHA1 and AHA2. Increased cytokinin signaling led to shortening of the meristem, similarly to an inducible relocalization of AHA2 fused to the glucocorticoid receptor. Finally, *Großeholz et al., 2022*, quantified the AHA2-GFP signal and demonstrated an increasing AHA2-GFP membrane abundance toward the maturation zone of *A. thaliana* root, which correlated with increased acidification of the root surface. Still, the observed patterns of proton fluxes along the root zones (*Weisenseel and Meyer, 1997*), and in particular the existence of the alkaline domain around the transition zone of the root, cannot be simply explained by the expression or abundance of AHA proton pumps.

Roots use the gravity vector as the reference for the direction of their growth. A general pattern was shown in various plant species: during the gravitropic response, ion fluxes change rapidly, leading to increased proton secretion on the upper and decreased secretion on the lower side, where it correlates with growth inhibition, and results in bending of the root (*Mulkey and Evans, 1981*; *Behrens et al., 1982*; *Iwabuchi et al., 1989*; *Collings et al., 1992*; *Weisenseel et al., 1992*). In *A.*

*thaliana*, the root surface and cell wall alkalinization of lower side upon gravistimulation has also been clearly demonstrated (*Monshausen et al., 2011*; *Shih et al., 2015*; *Barbez et al., 2017*), and it was shown that this alkalinization requires functional auxin transport (*Monshausen et al., 2011*) and the CNGC14 calcium channel acting downstream of auxin signaling (*Shih et al., 2015*). Alkalinization of the lower root side is thus triggered by the redirection of auxin flux toward the lateral root cap and epidermal cells on the lower side of the root (*Brunoud et al., 2012*). Analogously, apoplast alkalinization can be triggered by the application of auxin to the roots (*Evans et al., 1980*; *Lüthen and Böttger, 1988*; *Monshausen et al., 2011*; *Shih et al., 2015*; *Barbez et al., 2017*). The root surface pH increases almost immediately upon auxin application, and this process requires the activity of the CNGC14 calcium channel; in the mutant, surface alkalinization as well as the auxin-induced growth inhibition are delayed (*Shih et al., 2015*). The rapid root growth inhibition as well as the membrane depolarization and surface alkalinization triggered by auxin depend on the rapid branch of the TIR1/AFB signaling branch; auxin influx by AUX1 and the AFB1 auxin co-receptor paralogue play a prominent role in the rapid auxin response (*Fendrych et al., 2018*; *Dindas et al., 2018*; *Prigge et al., 2020*; *Serre et al., 2021*; *Li et al., 2021*; *Dubey et al., 2021*). The molecular mechanism of surface alkalinization is, however, not understood. It was proposed that auxin plays a dual role in regulation of root apoplastic pH: auxin alkalinizes the apoplast by the activity of an unknown ion channel downstream of the TIR1/AFB signaling, and, at the same time, auxin upregulates the activity of AHA H⁺ ATPases through TMK1 signaling (*Li et al., 2021*). It is not clear how auxin activates the TMK kinases. On the other hand, the connection between TIR1/AFB auxin signaling and activity of AHAs has been clarified: the auxin-induced SAUR proteins inhibit the PP2C-D clade of protein phosphatases that act as negative regulators of AHAs, leading to the activation of proton secretion. This pathway was studied mainly in shoots, but seems also to operate in roots, as the *Arabidopsis* lines with manipulated SAUR or PP2C-D expression show root phenotypes that are consistent with the expected changes in AHA activities (*Spartz et al., 2014*; *Ren et al., 2018*). In summary, the connection between auxin signaling, regulation of the activity of AHA H⁺ ATPases, and the longitudinal zonation of the root surface pH profile during normal growth and during gravitropic responses remains unclear.

Here, by employing an improved method for visualization and quantification of root surface pH, we focus on the molecular actors involved in the establishment of the longitudinal surface pH zonation. In particular, we show that the alkaline domain around the transition zone is not caused by the lack of AHA H⁺ ATPases activity. Further, we demonstrate that the alkaline domain at the transition zone is controlled by the components of the rapid auxin response pathway. Finally, we address the significance of the dynamic surface pH profile for the gravitropic response of the roots.

## Results

### The *Arabidopsis* root shows distinct acidic and alkaline root surface pH domains

To monitor the spatio-temporal dynamics of root ion fluxes and apoplastic pH in vertically growing roots of *A. thaliana*, we re-evaluated the available fluorescence staining methods to visualize pH in roots. Cell wall staining by 8-hydroxypyrene-1,3,6-trisulfonic acid trisodium salt (HPTS) (*Barbez et al., 2017*) was not satisfactory in our setup, due to the very high background signal and probably due to the absence of the optimal 458 nm excitation in our vertical stage spinning disk microscope (*Serre et al., 2021*). However, at the transition zone of the root, we could observe an alkaline domain on the root surface (*Figure 1—figure supplement 1a*). Further, we attempted to visualize root surface pH using Fluorescein Dextran and Oregon Green Dextran (*Figure 1—figure supplement 1a*) pH reporters (*Monshausen et al., 2011*; *Geilfus and Mühling, 2011*), but the results were not satisfactory due to an artifact when roots were imaged on solid medium (*Figure 1—figure supplement 1d*). We therefore searched for alternative pH-sensitive fluorescent dyes, and discovered Fluorescein-5-(and-6)-Sulfonic Acid, Trisodium Salt (FS) (Invitrogen F1130; *Seksek et al., 1995*; *Rosario and Rojas, 1986*) as an excellent reporter of root surface pH.

The $F_{488/405}$ excitation ratio of FS efficiently reported pH when dissolved in liquid or solid medium (*Figure 1a–d*, *Figure 1—figure supplement 1b,c*). The $F_{488/405}$ excitation ratio of FS is insensitive to the redox state of the medium but influenced by cation concentration (*Figure 1—figure supplement 1e*). FS allows to visualize pH of the root surface and the surrounding rhizosphere, without penetrating

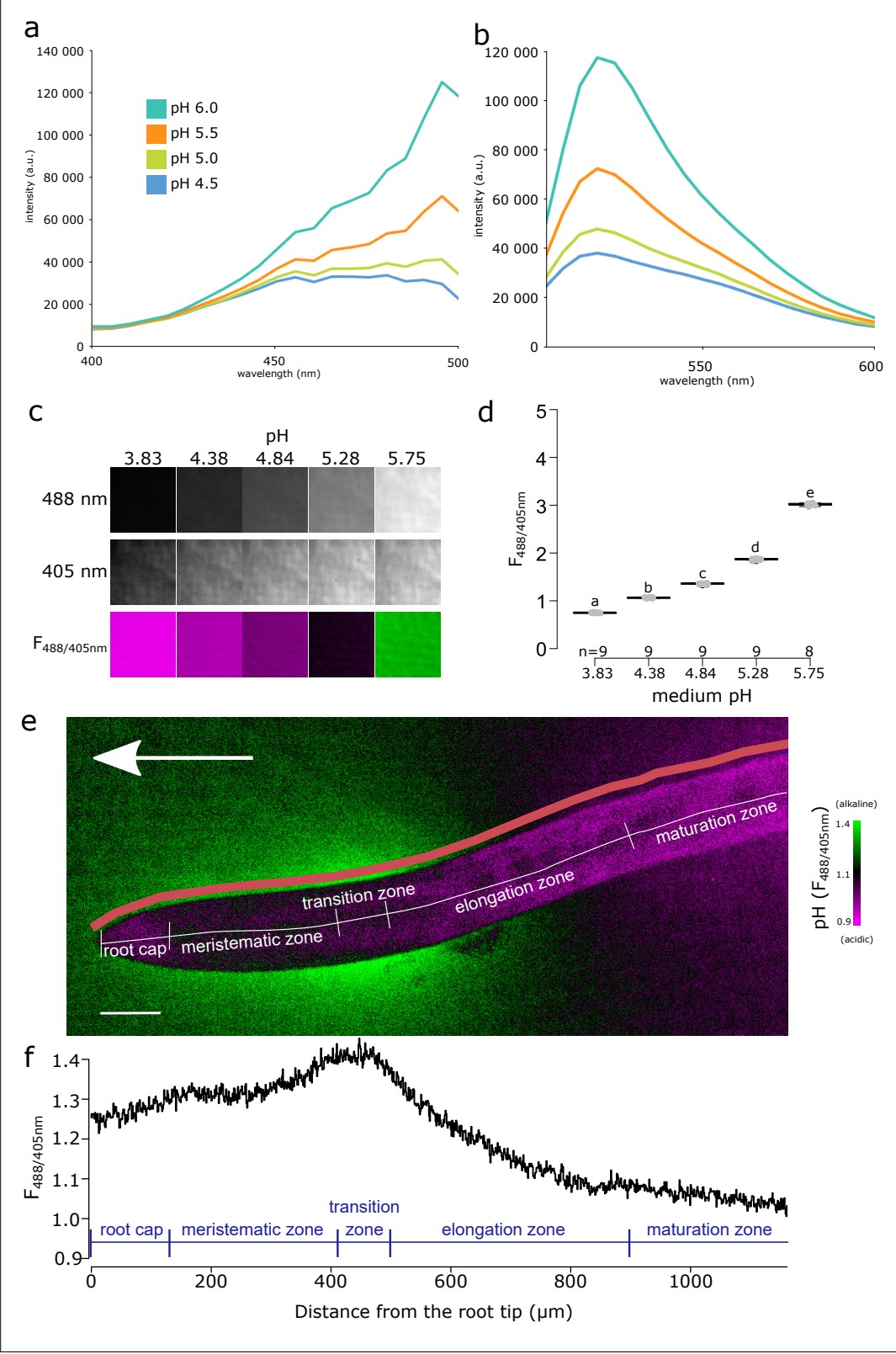

**Figure 1.** Fluorescein-5-(and-6)-Sulfonic Acid, Trisodium Salt (FS) dye reveals acidic and alkaline domains at the root root surface. (**a, b**) The pH dependence of the excitation (**a**) and emission (**b**) spectrum of FS in liquid plant growth medium. Excitation spectra were recorded at $\lambda$ Em = 520 nm, emission spectra were excited by $\lambda$ Ex = 488 nm. (**c**) FS fluorescence in solid agar medium at indicated pH values, fluorescence excited by 488 nm, 405 nm

*Figure 1 continued on next page*

*Figure 1 continued*

and the $F_{488/405}$ excitation ratio, LUT as in (**e**). (**d**) Quantification of the $F_{488/405}$ excitation ratio in (**c**). (**e**) *A. thaliana* root tip shows the alkaline and acidic surface pH domains, arrow indicates the gravity vector, scale bar = 50 μm. The color of the root itself does reflect pH, as the root itself is not stained by FS (see *Figure 1—figure supplement 1f*). The pink line shows the region in which F488/405 excitation ratio was plotted in (**f**). (**f**) The $F_{488/405}$ excitation ratio of FS along the longitudinal axis of the root. In (**e, f**), the typical root zones are depicted for illustration. The source data can be found in *Figure 1—source data 1*.

The online version of this article includes the following source data and figure supplement(s) for figure 1:

**Source data 1.** Data used for generating the graphs in the figure.

**Figure supplement 1.** Visualization of root surface pH.

**Figure supplement 1—source data 1.** Data used for generating the graphs in the figure.

the root tissues (*Figure 1e*). The pH imaging with FS highlighted the previously described relatively alkaline domain in the transition zone/early elongation zone of the root (*Monshausen et al., 2011*; *Staal et al., 2011*). In addition, we observed two relatively acidic pH domains, one located in the late elongation/root hair zone and the other in the proximity of the root tip (*Figure 1e*). To unbiasedly quantify the root surface pH, we developed a Python-based program to determine the $F_{488/405}$ ratio of the root surface (*Figure 1—figure supplement 1f*). The quantification highlighted the presence of the observed relatively alkaline and acidic domains (*Figure 1f*). Throughout the text, we will refer to these domains as alkaline and acidic domains, by which we mean values relatively acidic or relatively alkaline in comparison with the pH of the medium. The surface pH profile of roots shown by FS is in agreement with the data obtained using numerous electrode and microscopy measurements in several species including *A. thaliana* (*Zieschang et al., 1993*; *Staal et al., 2011*; *Monshausen et al., 1996*; *Weisenseel and Meyer, 1997*). The FS pH detection range thus reveals both the alkaline and acidic domains of the root surface and allows direct and dynamic visualization of proton concentration on the root surface and its close surroundings.

## The alkaline domain in the transition zone is not directly determined by AHA activation or localization

We first hypothesized that the observed spatial surface pH gradients might originate from gradients of AHA ATPase abundance or activity (*Großeholz et al., 2022*). We therefore immunolocalized endogenous AHAs using the antibody which recognizes multiple AHA ATPases paralogs in multiple plant species (Agrisera AS07260). To test the specificity of the antibody in *Arabidopsis* roots, we imaged Col-0 roots, *aha2-4* mutants lacking the dominant AHA2 paralogue, and lines inducibly expressing a dominant version of AHA2 (AHA2-d95-mScarlet). The antibody showed signal in the PM in Col-0 and *aha2-4* roots (*Figure 2—figure supplement 1a and b*). When we inducibly over-expressed the AHA2-d95-mScarlet, we detected colocalization of the mScarlet signal with the antibody signal (*Figure 2—figure supplement 1c*). These results indicate that the antibody recognized multiple plasma membrane localized AHAs in *Arabidopsis* roots. The immunolocalization of endogenous AHAs, however, didn't reveal any obvious absence of AHAs in the transition zone (*Figure 2a*, *Figure 2—figure supplement 1d*) that would explain the presence of the alkaline domain. To overcome the hypothetical lack of AHA activation in the TZ, we treated seedlings with fusicoccin (FC), a fungal toxin that stimulates the AHA's activity (*Ballio et al., 1964*) and thus increases proton efflux. FC lowered the surface pH of the root acidic domains but, surprisingly, did not affect the alkaline domain (*Figure 2b and c*). This partial acidification was correlated with the known FC-induced stimulation of root growth (*Figure 2—figure supplement 2a*).

To exclude an FC-independent regulation of AHAs in the TZ, we created *A. thaliana* lines inducibly expressing a fluorescently tagged hyperactive version of AHA2 (*Pacheco-Villalobos et al., 2016*) - AHA2-d95-mScarlet, and lines expressing an inhibitor of AHAs PP2CD1-mScarlet (*Ren et al., 2018*) under the control of the epidermal/cortex PIN2 promoter (*Figure 2d*). Expression of hyperactive AHA2 increased the acidic domain in the root tip, slightly reduced the alkaline domain, but did not prevent its formation (*Figure 2e and f*). This restricted acidification led to a tendency to stimulate root growth (*Figure 2—figure supplement 2b*).

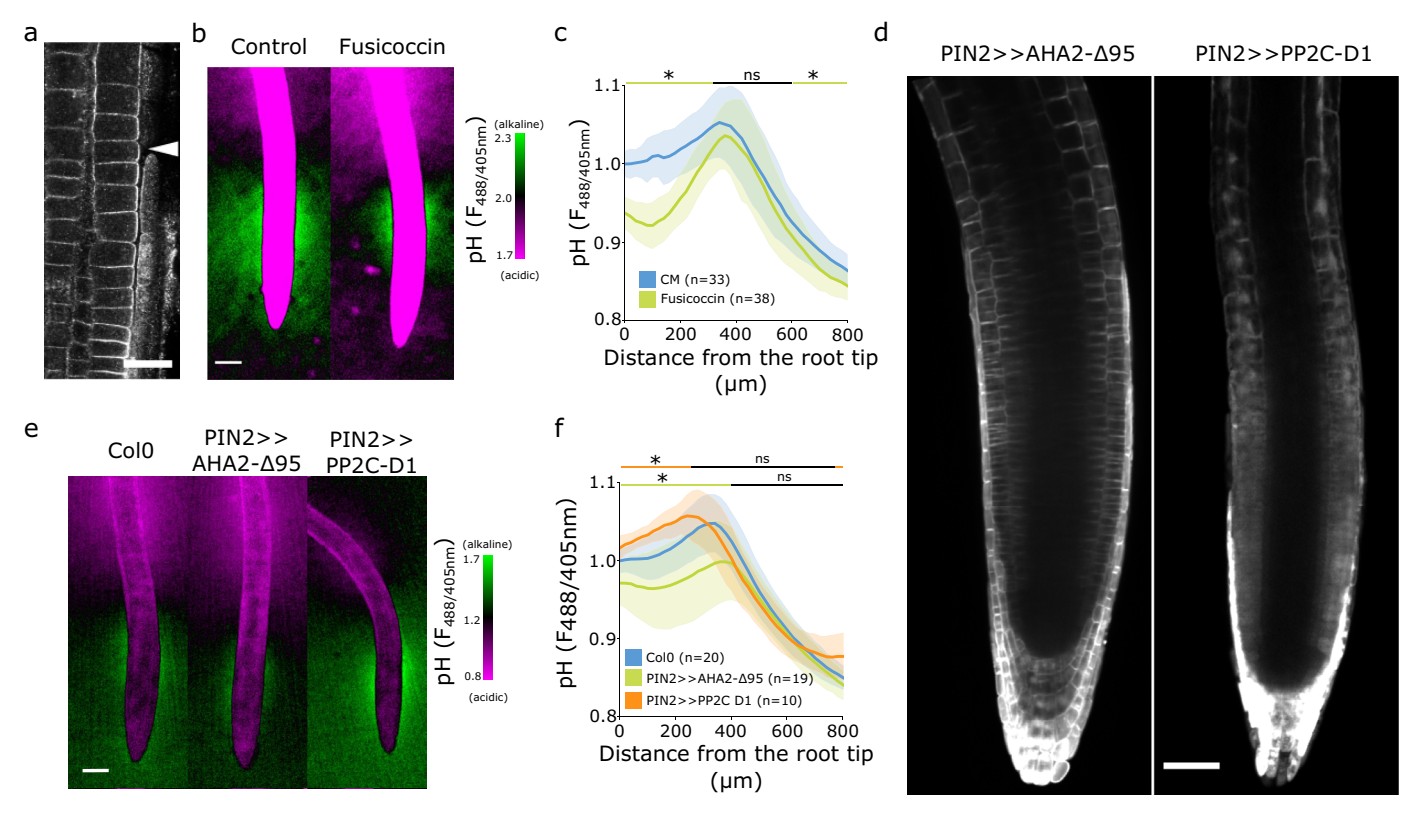

**Figure 2.** The alkaline domain does not directly depend on proton pump activity. (**a**) Immunostaining of AHA proton pumps in the Col-0 root transition zone (arrowhead). Scale bar = 20 µm. (**b,c**) Root surface pH visualized by FS of wild-type (WT) Col-0 seedlings treated with 0 µM (control) or 2 µM fusicoccin. (**b**) Representative image (scale bar = 100 µm) and (**c**) quantifications of FS F488/405 excitation ratio profile. (**d**) Image of root tips of the tissue-specific inducible lines PIN2 >>AHA2-Δ95-mScarlet and PIN2 >>PP2C-D1-mScarlet seedlings after 4 hr induction by 5 µM estradiol. Scale bar = 50 µm. (**e,f**) Root surface pH visualized by FS of WT Col-0 and induced PIN2 >>AHA2-Δ95-mScarlet and PIN2 >>PP2C-D1-mScarlet lines. (**e**) Representative images (scale bar = 100 µm), and (**f**) quantifications of FS F488/405 excitation ratio profile. For (**c**) and (**f**), the bars above the curves mark regions with non-significant (ns) and significant (*: p-value <0.05) statistical difference in comparison to control. The source data can be found in *Figure 2—source data 1*.

The online version of this article includes the following source data and figure supplement(s) for figure 2:

**Source data 1.** Data used for generating the graphs in the figure.

**Figure supplement 1.** Immunolocalization of AHAs.

**Figure supplement 2.** The influence of genetic manipulation of AHAs on root surface pH.

**Figure supplement 2—source data 1.** Data used for generating the graphs in the figure.

Inhibition of AHA activity by overexpression of PP2C-D1 raised the overall root tip surface pH, but did not prevent the formation of the alkaline domain in the TZ (*Figure 2e and f*). Alkalinization of the root tip surface correlated with a statistically insignificant reduction of root growth (*Figure 2—figure supplement 2b*). We further measured the root surface pH profile of *aha2* and *pp2c-d triple* mutants that showed decreased (*Haruta and Sussman, 2012*) and increased (*Ren et al., 2018*) AHA activity, respectively. Plants lacking the expression of PP2CDs showed a significantly acidified root tip surface, however, the alkaline halo remained unaffected (*Figure 2—figure supplement 2f and h*). On the other hand, plants lacking the expression of AHA2 were slightly impaired in the alkaline halo domain formation (*Figure 2—figure supplement 2f and h*). *aha2-4* roots were growing slower than Col-0 while *pp2c-d triple* root growth was stimulated (*Figure 2—figure supplement 2g*).

The manipulation of the AHA proton pump activity resulted in the expected outcome of influencing the overall root surface pH and root growth. However, the spatial organization of the root surface pH profile with the alkaline domain of the TZ does not appear to be simply controlled by the activation or

abundance of AHAs, as the alkaline domain cannot be fully removed by genetic or pharmacological modulation of the proton pumping activity.

## The establishment of the alkaline pH domain requires AUX1

It is well established that the application of auxin to roots causes an increase in apoplastic and root surface pH (*Monshausen et al., 2011*; *Shih et al., 2015*; *Li et al., 2021*). To investigate the effect of auxin application on surface pH profile and formation of the TZ alkaline domain, we analyzed surface pH in response to the native auxin indole-3-acetic acid (IAA). With the increasing IAA concentration, the alkaline domain pH rose and the domain expanded toward the elongation zone of the root, which resulted in disappearance of the acidic domain (*Figure 3a and b*). The extent of the surface alkalinization correlated with IAA-induced root growth inhibition; the response was detectable at 1 nM IAA and was saturated between 100 and 1000 nM IAA (*Figure 3c*). This implies that the entire root surface is capable of alkalinization upon external auxin treatment, and that the alkaline domain is the hotspot of auxin response.

We further analyzed the root surface pH response to IAA in line with altered AHA activity. The PIN2>>AHA2-d95 and PIN2>>PP2 CD1 (*Figure 2—figure supplement 2c and d*) as well as the *pp2c-d triple* mutant responded to IAA treatment by alkalinization of the root surface pH (*Figure 2—figure supplement 2h and i*). The alkalinization factors (a measure of IAA-induced alkalinization) were similar to Col-0 with the exception of the very root tip of *pp2c-d triple* which responded slightly less (*Figure 2—figure supplement 2e and j*). These results were correlated with control level IAA-induced root growth inhibition (*Figure 2—figure supplement 2b and g*). On the other hand, knockout mutation of AHA2 led to reduced IAA-induced root surface alkalinization (*Figure 2—figure supplement 2h, i and j*), but strong enough to observe an auxin-induced root growth inhibition (*Figure 2—figure supplement 2g*). The alkaline domain does not seem to be caused by the lack of proton efflux in the TZ, and the amplitude of the alkaline domain can be increased by auxin application. We therefore tested how auxin transport, perception, and response contribute to the spatial determination of root surface pH. First, we tested the mutant in the PIN2 auxin efflux carrier, in which the shootward auxin flux through the outer root tissues is perturbed (*Luschnig et al., 1998*; *Müller et al., 1998*). The *pin2* mutant root showed a reduced alkaline domain that was shifted toward the root tip (*Figure 3d and e*). Upon application of 10 nM IAA to *pin2* mutant, the alkaline domain position and amplitude were partially restored (*Figure 3—figure supplement 1a and b*), the *pin2* roots also responded to auxin by growth inhibition (*Figure 3—figure supplement 1c*). Upon treatment with 100 nM IAA, the alkaline domain and the overall IAA-induced root alkalinization were fully restored (*Figure 3d and f* and *Figure 3—figure supplement 1d and e*).

The auxin influx carrier AUX1 is essential for auxin uptake and transport during root gravitropism (*Swarup et al., 2005*; *Band et al., 2014*), and the null *aux1* mutant was shown to have a more acidic surface with an altered root tip pH profile (*Monshausen et al., 2011*). We determined the root surface pH profile in the *aux1* mutant, and found that its pH showed a gradual decrease from the tip toward the root hair zone. While the acidic zone in the elongation zone was comparable to control roots, *aux1* mutants displayed a complete absence of the TZ alkaline domain (*Figure 3d and e*). Application of 10 nM IAA on *aux1* mutants did not affect root elongation (*Figure 3—figure supplement 1c*) and did not trigger a root surface alkalinization in contrast to the control Col-0 (*Figure 3—figure supplement 1a and b*). However, at 100 nM IAA, when IAA diffusion compensates for AUX1 function (*Evans et al., 1994*), *aux1* mutant showed a clear IAA-induced root growth inhibition (*Figure 3—figure supplement 1e*) as well as an increased root surface pH beyond the TZ (*Figure 3d and f* and *Figure 3—figure supplement 1d*). The wild-type (WT)-like root surface pH profile was, however, not restored by IAA application, demonstrating that AUX1 is essential for creating the alkaline surface domain in the transition zone.

As the immunolocalization pattern of AHAs in the *aux1* mutant was comparable to the Col-0 control (*Figure 2—figure supplement 1d*), we treated the mutant with FC to investigate the activation status of AHAs in the *aux1* mutant. Interestingly, this treatment resulted in the establishment of a small alkaline domain also in the *aux1* mutant (*Figure 3—figure supplement 1f and h*), likely caused by AHA-mediated acidification in the root tip and the distal elongation zone that did not affect the TZ surface pH. The partial acidification was again correlated with a significant root growth stimulation (*Figure 3—figure supplement 1g*). This experiment showed that AHAs are not fully activated in *aux1*

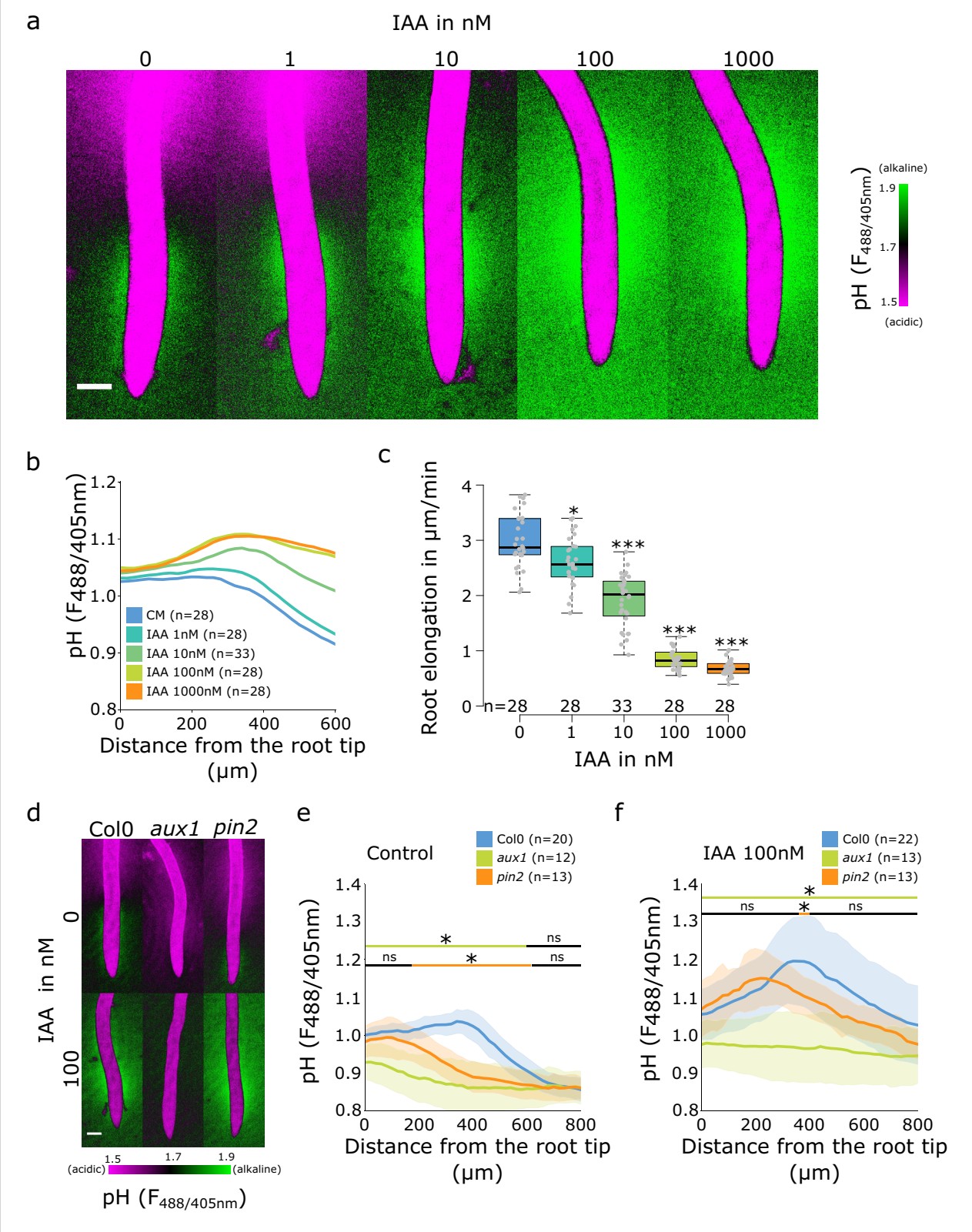

**Figure 3.** Auxin influx by AUX1 is essential for the initiation of the alkaline domain. (**a–c**) Surface pH correlates with growth rate-dose response of Col-0 roots to indole-3-acetic acid (IAA) auxin. (**a**) Representative images of root surface pH visualized by FS after 20 min IAA treatment. (**b**) Quantifications of FS F488/405 excitation ratio profile. (**c**) Root elongation rate (μm/min) measured over a 40 min period. (**d–f**) Root surface pH visualized by FS of Col-0, *aux1* and *pin2* mutants after 20 min treatment with 0 or 100 nM IAA. (**d**) Representative images. (**e**) Quantification of FS F488/405 excitation ratio profile

*Figure 3 continued on next page*

*Figure 3 continued*

in (**e**) control condition and (**f**) in response to 100 nM IAA. For (b,e,f), the bars above the curves mark regions with non-significant (ns) and significant (*: p-value <0.05) statistical difference in comparison to control. For (c), *: p-value <0.05, ***: p-value <0.0005. Scale bars = 100 µm. The source data can be found in *Figure 3—source data 1*.

The online version of this article includes the following source data and figure supplement(s) for figure 3:

**Source data 1.** Data used for generating the graphs in the figure.

**Figure supplement 1.** The role of auxin transport in regulation of root surface pH.

**Figure supplement 1—source data 1.** Data used for generating the graphs in the figure.

mutant and implies a specific surface pH response of the cells of the TZ. In summary, we have shown that the distinctive alkaline domain at the transition zone of the root depends on AUX1-mediated auxin influx. Without auxin influx, the root shows a linear acidification gradient from the root tip to the maturation zone.

## The components of the rapid auxin response pathway steer the transition zone root surface pH

Once in the cytoplasm, auxin triggers responses either through the canonical auxin signaling pathway or the auxin rapid response pathway (*Dubey et al., 2021*). We examined the involvement of these pathways in the establishment of the surface pH profile of the root. First, we investigated the components of the canonical auxin signaling by using the *tir1,afb2,3* (*tir triple*) mutant which lacks the expression of three of the six known auxin receptors (*Dharmasiri et al., 2005*). The *tir triple* mutant displayed a more acidic surface pH, particularly in the elongation and maturation zone (*Figure 4a and b*). However, it was still displaying an alkaline domain, albeit less pronounced than the control. In response to IAA, *tir triple* was critically impaired in whole root IAA-induced surface pH alkalinization (*Figure 4—figure supplement 1b and c*). As a result, *tir triple* was also impaired in the IAA-induced root growth inhibition (*Figure 4—figure supplement 1a*). These results confirm that the canonical auxin signaling is involved in the auxin-induced apoplastic alkalinization (*Li et al., 2021*) and also partially in the longitudinal surface pH zonation. Next, to test the role of the TMK1-ABP1 apoplastic auxin perception pathway (*Li et al., 2021*; *Friml et al., 2022*) in the establishment of root surface pH profile, we analyzed the root surface pH profiles of *tmk1*, *tmk4*, and *abp1* mutants. The surface pH profile of the mutants was, however, comparable to the Col-0 controls (*Figure 4—figure supplement 1d, e and f*).

We further explored the role of the recently discovered molecular actors of the auxin rapid response - AFB1 and CNGC14. AFB1 is the paralogue of the TIR1 receptor; AFB1 has been shown to be crucial for the rapid growth inhibition upon auxin treatment, rapid auxin-induced membrane depolarization, and for the early response to gravistimulation (*Prigge et al., 2020*; *Serre et al., 2021*). Similarly, the mutant in the calcium channel CNGC14 lacks the auxin-induced calcium transient, membrane depolarization, and shows a delay in the early gravitropic response (*Shih et al., 2015*; *Dindas et al., 2018*). We analyzed the root surface pH of both mutants and found a more acidic root surface with a critically flat alkaline domain in the *afb1* mutant (*Figure 4c and d* and *Figure 4—figure supplement 2a*) as well as in the *cngc14* mutant (*Figure 4e and f* and *Figure 4—figure supplement 2b*). We confirmed the similar result in additional alleles of *afb1* (*Figure 4—figure supplement 2c*) and *cngc14* mutants (*Figure 4—figure supplement 2d*). This shows that these proteins, apart from controlling the rapid auxin response, steer the pH profile of the root during normal steady-state growth by controlling the alkaline domain in the transition zone. The lack of the alkaline domain was not caused by mislocalization or absence of AUX1 protein in the *afb1* or *cngc14* mutants (*Figure 4—figure supplement 2e*). Similarly to the *aux1* mutant, the localization of AHA H+ ATPases in both mutants was not obviously different from the Col-0 control (*Figure 2—figure supplement 1d*). We further tested how the *afb1* and *cngc14* mutants respond to IAA and found an overall alkalinization of the root surface with a partial rescue of the alkaline domain (*Figure 4d and f*) and partially impaired growth inhibition (*Figure 4—figure supplement 2f–h*).

What makes the TZ zone surface pH different from the other domains of the roots? A plausible explanation is the localization of relevant molecular components in this region of the root. The auxin transporters AUX1 and PIN2 have been shown to localize to the lateral root cap and epidermis (*Müller*

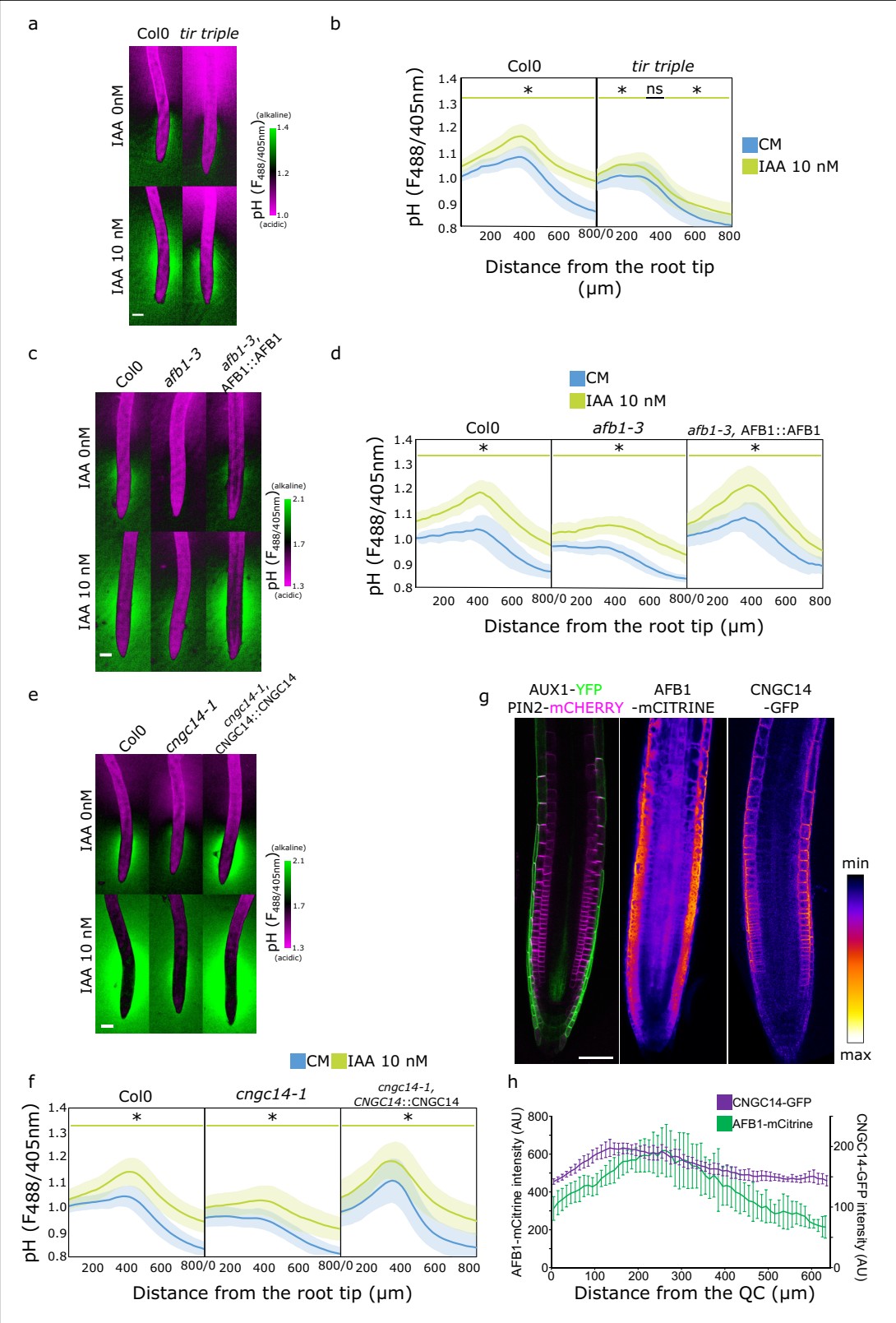

**Figure 4.** Rapid auxin signaling steers root surface pH. (**a, b**) Root surface pH of Col-0 and *tir triple* mutant after 20 min treatment with 0 or 10 nM indole-3-acetic acid (IAA). (**a**) Representative images of pH visualization by FS and (**b**) quantification of FS F488/405 excitation ratio profile. (**c–d**) Root surface pH of Col-0, *afb1-3* mutant, and AFB1::AFB1-mCitrine/*afb1-3* complemented line after 20 min treatment with 0 or 10 nM IAA. (**c**) Representative images of pH visualization by FS and (**d**) quantification of FS F488/405 excitation ratio. (**e–f**) Root surface pH of Col-0, *cngc14-1* mutant, and

*Figure 4 continued on next page*

*Figure 4 continued*

CNGC14::CNGC14-GFP/*cngc14-1* complemented line after 20 min treatment with 0 or 10 nM IAA. (**e**) Representative images of pH visualization by FS and (**f**) quantification of FS F488/405 excitation ratio. (**g**) Localization of AUX1, PIN2, AFB1, and CNGC14 proteins driven by their respective native promoters. For (b,d,f), the bars above the curves mark regions with non-significant (ns) and significant (*: p-value <0.05) statistical difference in comparison to control. Scale bars = 100 μm (**a,c,e**) or 50 μm (**g**). (**h**) Quantification of AFB1-mCitrine and CNGC14-GFP signal intensity in the root epidermis, n=9 (AFB1) and n=15 (CNGC14) roots; error bars = st.dev. The source data can be found in *Figure 4—source data 1*.

The online version of this article includes the following source data and figure supplement(s) for figure 4:

**Source data 1.** Data used for generating the graphs in the figure.

**Figure supplement 1.** The role of TMK-ABP1 signaling in regulation of root surface pH.

**Figure supplement 1—source data 1.** Data used for generating the graphs in the figure.

**Figure supplement 2.** Root surface pH and growth of additional *afb1* and *cngc14* mutant alleles.

**Figure supplement 2—source data 1.** Data used for generating the graphs in the figure.

*et al., 1998*; *Swarup et al., 2004*, *Figure 4g*). *Prigge et al., 2020*, showed that AFB1 is expressed in the root tip and we could see an enrichment of the protein in the root epidermis (*Figure 4g*). To determine the localization of CNGC14, we expressed its fluorescently C-terminally tagged version under the control of its native promoter. The expression of CNGC14 fusion protein was rather weak, and localized to the PM of root epidermal cells, with an enrichment in the transition zone (*Figure 4g and h*). The expression of AFB1-mCitrine and CNGC14-GFP in the respective mutants recovered the presence of the TZ alkaline domain (*Figure 4c, d, e and f* and *Figure 4—figure supplement 2a, b and d*). Interestingly, the expression of CNGC14-GFP in *cngc14* caused an exaggerated TZ alkaline domain (*Figure 4e and f* and *Figure 4—figure supplement 2b and d*), which underlines the importance of the CNGC14 calcium channel in the establishment of root surface pH profile.

These results show that the signaling components so far associated with the rapid auxin response are expressed in the root epidermis and contribute to the longitudinal zonation of root surface pH profile. The alkaline domain of the TZ represents a site of a constant rapid auxin response that is triggered by the internal auxin fluxes mediated by AUX1 and PIN2 transporters. On the other hand, the overall increase of root surface pH upon auxin treatment can be attributed to the TIR1 canonical signaling pathway.

## The AUX1-AFB1-CNGC14 module facilitates root navigation

It was shown that auxin induces alkalinization of the apoplastic pH and that this alkalinization correlates and is required for the inhibition of root growth. The same process occurs during the gravitropic response - the lower root side responds to the internal auxin by alkalinization, resulting in root bending (*Monshausen et al., 2011*; *Shih et al., 2015*; *Barbez et al., 2017*). We reanalyzed the root surface pH using our pH imaging and quantifications, as it enables us to monitor alkalinization, as well as acidification of the root surface. Upon gravistimulation, the TZ alkaline domain on the lower side of the root increased rapidly, while on the upper side, the alkaline domain diminished and the root surface acidified (*Figure 5a*, *Video 1*). This led to a gradient of surface pH across the root that was established within 5 min of the gravistimulation which correlated with initiation of bending of the root (*Figure 5a*). The disappearance of the alkaline halo on the upper root surface indicates that the TZ domain might act as a zone of stalled growth which is rapidly activated upon gravistimulation. We further analyzed the gravitropic responses of the mutants with altered root surface pH zonation. As expected, the agravitropic *aux1* mutant did not create a pH gradient across the root and did not bend (*Figure 5b*). Manipulation of the AHA activity by the application of FC or by genetic means did not prevent the formation of a gradient of surface pH and rapid bending of the root (*Figure 5—figure supplement 1a and b*). Finally, the *afb1* and *cngc14* mutants showed a slower gravitropic response, as reported before (*Serre et al., 2021*; *Shih et al., 2015*; *Figure 5c and d*). As both mutants have a diminished alkaline domain in the TZ, upon gravistimulation, this zone could not rapidly react to change in auxin fluxes; instead, a shallow gradient of surface pH slowly develops in the mutants (*Figure 5c and d*).

We noticed that the alkaline domain showed dynamic fluctuations during vertical growth of the WT root (*Figure 5e*, *Video 2*), similarly to what was described by *Monshausen et al., 2011*. In the *aux1*, *afb1*, and *cngc14* mutants, the alkaline domain was absent or less prominent during vertical growth (*Video 3*). This result and the fact that the components of the rapid auxin response pathway

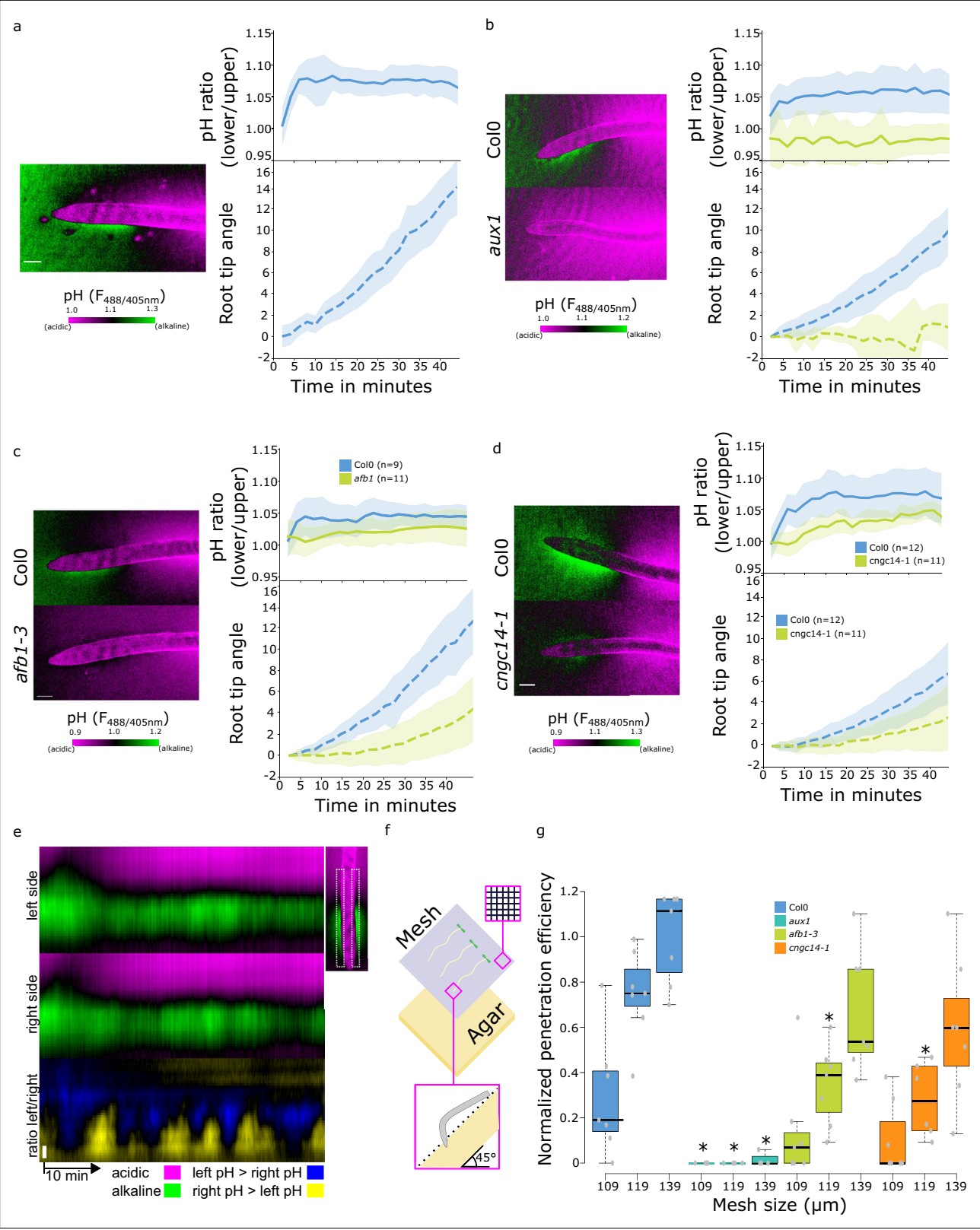

**Figure 5.** Rapid auxin signaling pathway is required for surface alkalinization during rapid gravitropic responses. (**a–c**) Surface pH dynamics and root tip bending angle during gravitropic response in (**a**) Col-0, (**b**) Col-0 and *aux1*, (**c**) Col-0 and *afb1-3*, (**d**) Col-0 and *cngc14-1* lines. A representative image of pH visualization by FS, quantification of the FS F488/405 excitation ratio of lower/upper root transition zones, and root tip angle dynamics over time are shown for each line. Representative images were taken 40 min after gravistimulation. (**e**) Root surface pH oscillations in vertically growing Col-0 roots.

*Figure 5 continued on next page*

*Figure 5 continued*

The FS F488/405 excitation ratio images for left and right root sides and their left/right ratio are shown. (**f,g**) Root tip penetration test in 45° inclined media covered with mesh of different pore sizes (109, 119, and 139 µm). (**f**) Schematics of the experimental setup. (**g**) Quantitation of Col-0, *aux1*, *afb1-3*, and *cngc14-1* mesh penetration efficiencies. For (g), statistical differences with p-value <0.05 indicated by *. All scale bars = 100 µm. The source data can be found in *Figure 5—source data 1*.

The online version of this article includes the following source data and figure supplement(s) for figure 5:

**Source data 1.** Data used for generating the graphs in the figure.

**Figure supplement 1.** Surface pH dynamics and root tip bending angle during gravitropic responses of indicated lines and treatments.

**Figure supplement 1—source data 1.** Data used for generating the graphs in the figure.

are involved in the formation of the alkaline domain indicates that, in this zone, the cells continuously rapidly respond to the internal auxin fluxes, and that this pathway is active not only upon gravistimulation. We hypothesized that the significance of this process is to constantly correct the growth rate fluctuations and to enable the root to quickly regulate the growth direction of the root tip. This would be particularly relevant in the soil environment, where the root tip penetrates the soil particles and constantly corrects growth direction using the gravity vector. To test this hypothesis, we tested the ability of WT and *aux1*, *afb1*, and *cngc14* mutants to navigate through artificial obstacles, approximated by tilted agar plates covered with a rectangular nylon grid. We scored the efficiency of the roots in penetrating the grids with varying pore sizes (*Figure 5f*). WT plants penetrated the meshes more efficiently as the pore size increased, whereas the agravitropic *aux1* mutant failed to penetrate in all conditions. Interestingly, both the *afb1* and *cngc14* mutants showed a significant decrease in penetration efficiency compared to the WT (*Figure 5g*).

In summary, the alkaline surface pH domain originates from a constant rapid auxin response driven by the AUX1-AFB1-CNGC14 module. This response enables the roots to rapidly react to gravity vector changes and to adjust the root tip growth direction.

## Discussion

In this work, we present a new method to determine the root surface pH of *A. thaliana*. Its advantage is the sensitivity of FS in lower pH range that enables a spectacular visualization of root surface pH, including the acidic domain surrounding the late elongation and maturation zones of roots, which was not detected with the previously published methods (*Monshausen et al., 2011*; *Shih et al., 2015*). FS is an inexpensive dye which can be excited by the common 405 and 488 nm laser lines. In this work, we avoided measuring absolute pH values; instead, we determined the relative pH compared to internal controls observed on the same medium. Absolute pH determination would be possible, if extensive pH calibration was performed during each imaging session. This approach would be very laborious, and, in addition, absolute fluorescence intensities might depend on the thickness of the layer of

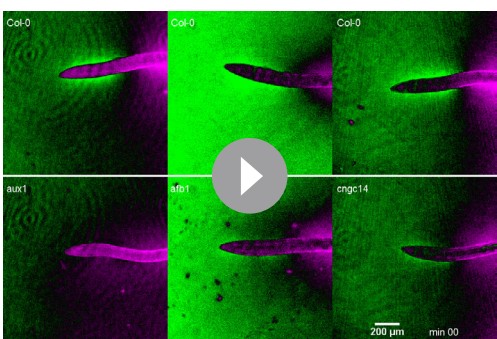

**Video 1.** Compilation of gravitropic responses of indicated mutant lines and controls.

https://elifesciences.org/articles/85193/figures#video1

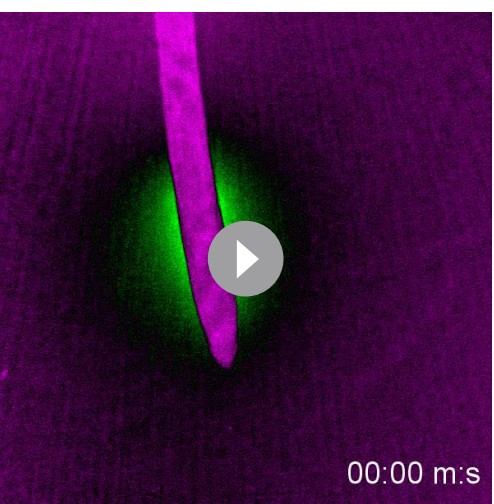

**Video 2.** High-resolution dynamics of root surface pH in Col-0 root.

https://elifesciences.org/articles/85193/figures#video2

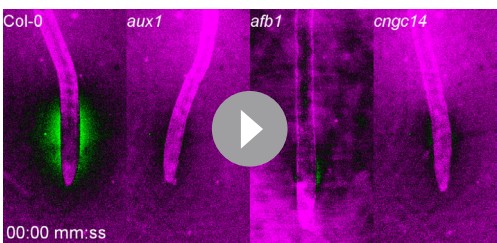

**Video 3.** High-resolution dynamics of root surface pH in Col-0, aux1, afb1, and cngc14 roots. The movie is assembled from individual movies, the lookup table is set identically for all genotypes.

https://elifesciences.org/articles/85193/figures#video3

medium and many other parameters. Therefore, we recommend using FS to determine relative pH changes compared to internal controls, that is WT plants or mock controls imaged together with mutants or treated roots. We showed that apart from pH, FS can also react to other ions, however, given that the surface pH profile visualized by FS staining is in full accordance with the previously published works (*Weisenseel and Meyer, 1997*; *Weisenseel et al., 1979*; *Björkman and Leopold, 1987*; *Behrens et al., 1982*; *Iwabuchi et al., 1989*; *Staal et al., 2011*), we conclude that in the growth media, FS reflects primarily the root surface pH.

According to the acid growth theory, acidic pH enables cell wall extension (*Rayle and Cleland, 1992*; *Hager, 2003*). We observed the lowest pH values in the very tip of the root and in the maturation zone where cells do not elongate. On the other hand, the alkaline domain partially covers the early elongation zone. The surface pH therefore does not fully correlate with the local growth rates (*Beemster and Baskin, 1998*; *Shih et al., 2015*). In addition, it remains to be resolved to what extent the surface pH correlates with the pH of the cell walls and the apoplastic space of the root epidermal cells, as previous work has reported a correlation between cell length and cell wall pH (*Barbez et al., 2017*; *Moreau et al., 2022*). It is possible that rather than driving cell elongation, the acidic domain in the maturation zone plays a role in nutrient acquisition and absorption by the root hairs (*Martín-Barranco et al., 2021*). The cytokinin-mediated cell wall stiffening may become the dominant growth-regulating mechanism in this root zone, leading to growth cessation despite low extracellular pH (*Liu et al., 2022*).

Based on genetic and pharmacological experiments we concluded that the longitudinal surface pH zonation is not based solely on the activity and abundance of PM AHA H$^+$ ATPases, but instead, the alkaline domain in the transition zone is driven by AUX1-mediated auxin influx and AFB1 signaling. The surface pH might be determined solely by AHA activity in the specific case of the *aux1* mutant, where the pH progressively decreases in the shootward direction, and might then reflect the AHA activity gradient that is determined by the abundance of the proton pumps and their regulation by brassinosteroid signaling, as suggested by *Großeholz et al., 2022*. Interestingly, even in the *aux1* mutant, a residual AHA-independent alkaline domain is present and can be visualized when AHAs are hyperactivated by the application of FC.

In agreement with these results, the ectopic AHA ATPases activation also cannot prevent the auxin-induced growth inhibition and the gravitropic bending of roots. Inhibition of AHA activity led to slower growth, a tendency of slower gravitropic response, and the *aha2-4* mutation caused a reduced auxin-induced surface alkalinization. In addition to controlling membrane potential and apoplastic pH, AHA activity influences auxin signaling by regulating auxin influx and diffusion into cells (*Rubery and Sheldrake, 1973*; *Yang et al., 2006*). These results might thus be caused by the reduced proton motive force that in turn might interfere with IAA uptake into the cells.

What is the cause of the alkaline domain at the transition zone, if not the inhibition of AHA ATPase activity? It was shown that the auxin-induced apoplast and root surface alkalinization is accompanied by a decrease in cytosolic pH (*Monshausen et al., 2011*; *Li et al., 2021*), indicating that the alkaline domain visualized by the FS is caused by a proton influx into cells. The phenotype of the *aux1* mutant hints to the possibility that the AUX1-mediated auxin symport with protons (*Lomax et al., 1985*) might cause the observed root surface alkalinization. Also supporting this hypothesis, the transition zone of maize roots is the site of the highest auxin influx, as measured by an auxin-specific electrode (*Mancuso et al., 2005*). On the other hand, the *aux1* mutant was capable of surface alkalinization when higher concentrations of IAA were added, demonstrating that the AUX1-mediated influx per se is not required for surface alkalinization. Further, the *afb1* mutants show a defect in the alkaline domain formation, which favors the explanation that the auxin-induced alkalinization is triggered from the AFB1 receptor. Given that the mutant in CNGC14 is defective in establishing root surface pH

profile, auxin-induced alkalinization and auxin-induced calcium influx (*Shih et al., 2015*), and calcium influx and pH changes are tightly coupled (*Behera et al., 2018*), we conclude that CNGC14 functions downstream of AFB1. Similarly, *Li et al., 2021*, suggested that auxin-induced alkalinization of the apoplast is mediated by a yet unknown mechanism. It is intriguing to speculate that the signaling pathway might operate via the adenylate cyclase activity of TIR1/AFB auxin receptors (*Qi et al., 2022*).

We show that the rapid auxin response pathway, so far connected with the reaction to auxin application (*Fendrych et al., 2018*; *Li et al., 2021*; *Monshausen et al., 2011*) or the gravitropic response (*Serre et al., 2021*; *Shih et al., 2015*), operates constantly in the growing *A. thaliana* root. Even though the other TIR1/AFB co-receptors partially contribute to the longitudinal surface pH zonation, AFB1 plays the most important role. The other TIR1/AFB receptors seem to be more important for the overall pH rise upon the application of IAA, in agreement with the results of *Li et al., 2021*. The localization of the alkaline zone seems to be determined by the intersection of the PIN2-mediated shootward auxin flux (*Luschnig et al., 1998*; *Müller et al., 1998*), localization of the AUX1-mediated auxin influx (*Swarup et al., 2005*; *Band et al., 2014*), and the enrichment of the AFB1 and CNGC14 proteins. We propose that the dynamic nature of the surface alkalinization observed by us and others (*Monshausen et al., 2011*; *Shih et al., 2015*) results from the interaction of the intensity of the auxin flux and the constant rapid response that occurs in the alkaline domain of the root tip. In addition, the auxin-induced alkalinization is constantly being counterbalanced by the TMK-mediated AHA ATPase activation and apoplast acidification (*Li et al., 2021*; *Friml et al., 2022*; *Lin et al., 2021*).

Upon gravistimulation, the alkaline domain on the lower side of the root increased rapidly, while on the upper side, the alkaline domain diminished and the root surface acidified, which is consistent with previous pH measurements done by electrodes (*Zieschang et al., 1993*; *Monshausen et al., 1996*), indicating that the alkaline domain corresponds to a zone of a stalled cell elongation that can be rapidly regulated upon gravistimulation. In addition to the gene expression changes that occur during obstacle avoidance (*Jacobsen et al., 2021*) and the FERONIA-mediated mechanoperception (*Shih et al., 2014*), the rapid auxin response module in the root decreases the reaction time of the root to the changes of root growth direction, and thus increases the efficiency of root soil penetration.

## Materials and methods
### Plant material used
We used the Col-0 ecotype and the following mutant and transgenic lines. The PIN2>>AHA2delta95, PIN2>>PP2C-D1 originate from this study, see Molecular cloning. Further, we used the *aha2-4* (SALK_082786), *pp2c-d2/5/6* triple mutant (*Ren et al., 2018*) (pp2c-d2 WsDsLOX493G12, pp2c-d5GABI_330E08, and pp2c-d6 SAIL_171H03), *aux1-100* (SALK_020355), *pin2* (NASC_N16706). The *afb1-3* and AFB1::AFB1-mCitrine in *afb1-3* originate from *Prigge et al., 2020*. The *afb1-1s* (SALK_144884C) was genotyped using the following primers SALK_LBb1.3, AACGGAAGACTA GGAAGCGAG, GCAACAGCTTCAAGACCTTTG. The *cngc14-1* (SALK_206460) was genotyped by SALK_LBb1.3, CACCTGCTTGTAAAGCAAAGG, TCGGAACAATTGGCAGAATAC; *cngc14-2* (wiscD-sLox437E09) by WisDsLox_LBP (AACGTCCGCAATGTGTTATTAAGTTGTC), TGTTTCACGTAAAGTC AAACCC, TAAGAATCCAAGTGGCCACAC. The CNGC14::CNGC14-GFP (see Molecular cloning) was transformed into the *cngc14-1* and *cngc14-2* homozygous lines. The *tir triple* is the *tir1,afb2,3* mutant (*Dharmasiri et al., 2005*). We used the *tmk1-1* (SALK_016360), *tmk4-1* (GABI_348E01) (*Li et al., 2021*), and the *abp1-TD1* (SK21825) (*Gao et al., 2015*). AUX1::AUX1-YFP (*Swarup et al., 2004*) was introduced into *afb1-3* and *cngc14-2* by crossing. AUX1::AUX1-YFP was crossed with PIN::PIN2-mCherry (*Retzer et al., 2019*).

Seeds were surface-sterilized by chlorine gas for 2 hr. Seeds were sown on 1% (wt/vol) plant agar (Duchefa) with ½ Murashige Skoog salts (MS, Duchefa), 1% (wt/vol) sucrose, adjusted to pH 5.8 with KOH, and stratified for 2 days at 4°C. Seedlings were vertically grown for 5 days in a growth chamber at 23°C by day (16 hr), 18°C by night (8 hr), 60% humidity, and light intensity of 120 µmol photons/m$^2$/s.

### Molecular cloning
The tissue-specific estradiol-inducible lines were prepared as follows. For PIN2>>AHA2delta95, we cloned the CDS 1-886 of AHA2 (AT4G30190) lacking the last 95 amino acids as in *Pacheco-Villalobos*

*et al., 2016*, and fused mScarlet-I (*Bindels et al., 2017*) to the C-terminal part. For PIN2>>PP2C-D1, we cloned the CDS of PP2C-D1 (AT5G02760) and fused mScarlet-I to the C-terminal part. Both constructs were cloned downstream of the 4xLexA Operon fused to CaMV 35S minimal promoter (*Sarrion-Perdigones et al., 2013*) and the transcriptional units were terminated by the 35S terminator and cloned into the alpha1-3 vectors (*Dusek et al., 2020*). XVE (*Zuo et al., 2000*) was cloned under the control of the PIN2 promoter (1.4 kb upstream of the AT5G57090), terminated by the RuBisCo terminator from *Pisum sativum* and the resulting transcriptional unit was cloned into alpha1-1 vector. The alpha transcriptional units were then interspaced with matrix attachment regions (*Dusek et al., 2020*), combined with a Basta resistance cassette and introduced into the pDGB3omega1 binary vector (*Sarrion-Perdigones et al., 2013*). The CNGC14::CNGC14-GFP construct contains the CNCG14 promoter (1.5 kb upstream of the start codon) and the CNGC14 coding sequence (the AT2G24610.1 splice variant) with the GFP fused to the C-terminus and 35S terminator. This transcriptional unit was combined with a kanamycin resistance cassette and FastRed marker for rapid selection of transgenic seeds into the pDGB3omega1 binary vector. All cloning steps were performed using the GoldenBraid methodology (*Sarrion-Perdigones et al., 2013*; https://gbcloning.upv.es/).

Col-0 ecotype (PIN2>>AHA2delta95, PIN2>>PP2C-D1) or *cngc14-1* and *cngc14-2* lines (CNGC14-mVenus) were transformed using the floral dip method (*Clough and Bent, 1998*).

## Pharmacological treatments and dyes

Treatments were prepared using the following chemicals: IAA (10 mM stock in ethanol; Sigma-Aldrich), FC (1 mM stock in ethanol; Sigma-Aldrich), estradiol (20 mM stock in DMSO; Sigma-Aldrich). Fluorescein Dextran 10,000 MW, Anionic (D1821, Thermo Fisher) - 10 mg/ml stock in miliQ $H_2O$, final concentration in media is 29 µg/ml. Oregon Green 488 Dextran, 10,000 MW (D7170, Invitrogen) - 1 mg/ml stock in miliQ $H_2O$, final concentration in media is 2.5 µg/ml. HPTS (H1529, Sigma-Aldrich), 100 mM stock in $H_2O$, final concentration in media is 1 mM. FS (F1130, Thermo Fisher), 100 or 50 mM stock in $H_2O$, final concentration in media is 50 µM.

Induction of PIN2>>AHA2delta95 and PIN2>>PP2C-D1 lines was done by incubating 5-day-old seedlings for 2.5 hr in 1/2 MS MES buffered media (g/l), 1% sucrose, pH 5.8 containing 2 µM estradiol before experiments.

## Measurement of FS excitation and emission spectra

For measurement of fluorescent spectra of FS, ½ MS media with 1% sucrose were prepared with pH 4.5, 5.0, 5.5, 6.0 (adjusted with KOH). FS stock (50 mM in $H_2O$) was added to the media to achieve a final concentration of 50 µM. For the measurement, 200 µl of this media was pipetted into a 96-well plate, five wells per each pH value and the spectras were measured using Spark multimode microplate reader (Tecan). The values for each pH were averaged and plotted.

## Measurement of FS range and effects of salts and redox status on FS signal

To demonstrate the wide pH-reporting range of FS (*Figure 1c*), citric acid and sodium citrate were mixed in various ratios and water was added to obtain citrate buffers with a range of pH values. The pH of citrate buffers was measured, the buffers were mixed with agar (0.8% final concentration), boiled and FS was added (50 µM final concentration). The mixture was poured into a Petri dish. After cooling down, slabs of the solidified buffers were cut out, and placed into imaging chambers.

To measure the effects of ions and redox status (according to *Martinière et al., 2013*) on FS fluorescence (*Figure 1—figure supplement 1e*), MES buffer (1 g/l) was prepared and boiled with agar (1% final concentration), supplied with salts, $H_2O_2$, DTT, and FS was added (50 µM final concentration). The mixture was poured into a Petri dish. After cooling down, slabs of the solidified buffers were cut out, placed into imaging chambers and imaged using a spinning disk microscope, see Imaging of root surface pH. The signal intensity was measured in both 405 and 488 nm excitation channels and $F_{488/405}$ ratio was calculated.

## Imaging of root surface pH

For imaging of surface pH, 5-day-old seedlings were transferred to unbuffered ½ MS, 1% sucrose, pH 5.7 (adjusted with KOH) containing 50 µM of FS +/-treatments and allowed for recovery 25 min

before imaging. The medium was prepared as follows: For 100 ml of medium, 1 g of plant agar and 1 g sucrose (wt/vol) were added to 100 ml of unbuffered ½ MS in a 250 ml reagent bottle. The media was boiled until the agar was completely dissolved. Note: we always prepared a fresh solid medium for each experiment from a stock of liquid ½ MS, sucrose and agar, as re-boiling the medium might affect the results. The medium was cooled to a temperature of 45–50°C and FS was added to achieve 50 µM concentration. This pre-solution was then divided into two equal volumes to make control and treatment medium.

For the gravitropic experiments, the seedlings were rotated ±90°. Only roots with a starting angle of 90°±10° were selected for analysis to obtain homogeneous gravitropic stimulations.

Surface pH imaging was performed using a vertical stage Zeiss Axio Observer 7 with Zeiss Plan-Apochromat 10×/0.8, coupled to a Yokogawa CSU-W1-T2 spinning disk unit with 50 µm pinholes and equipped with a VS401 HOM1000 excitation light homogenizer (Visitron Systems). Images were acquired using the VisiView software (Visitron Systems).

FS was sequentially excited with a 488 and 405 nm laser and the emission was filtered by a 500–550 nm bandpass filter. Signal was detected using a PRIME-95B Back-Illuminated sCMOS Camera (1200×1200 pixels; Photometrics). The Flat field correction mode was used for image acquisition.

For vertical growth experiments, seedlings were imaged every 10 min for 30 min. The profiles shown correspond to the first time frame while the root elongation was calculated over the whole experiment. Imaging to observe oscillations of the root surface pH was conducted by imaging every 5 s.

For gravitropic experiments, seedlings were imaged using the sandwich method (https://doi.org/10.1017/qpb.2022.4) every 2 min for 42 min. The profile shown corresponds to the last time frame except specified otherwise. The angles were quantified over the entire duration of the experiment.

The images in *Figure 4g* were acquired using the Zeiss LSM 880 confocal microscope, with a 25×/0.8 water immersion objective using settings appropriate for the respective fluorophores.

## Immunolabeling of AHAs

Whole mount immunolocalization of 5-day-old *A. thaliana* seedlings was performed as described previously in *Sauer et al., 2006*, with the following modifications. The protocol was adapted to the InSituPro VS liquid-handling robot (Intavis AG, Germany). Prior to immunolocalization, seedlings were fixed 1 hr with 4% paraformaldehyde dissolved in MTSB (50 mM PIPES, 5 mM EGTA, 5 mM MgSO$_4$·7H$_2$O pH 7, adjusted with KOH) at room temperature. In the robot, procedure started with 5×15 min washes with MTSB-T (MTSB+0.01% Triton X-100) then the cell wall was digested by 30 min treatment at 37°C with 0.05% Pectolyase Y-23 supplemented with 0.4 M mannitol in MTSB-T, followed by 2×30 min membrane permeation with 10% DMSO and 3% Igepal in MTSB-T. The samples were blocked 1 hr with BSA (blocking solution: 2% BSA in MTSB-T) and incubated 4 hr at 37°C with primary (antiAHA rabbit antibody, Agrisera AS07260, RRID:AB_1031584) and 3 hr at 37°C with secondary antibody (Alexa Fluor 555 goat, anti-rabbit, Abcam ab150078, RRID:AB_2722519). The antibodies were diluted in 2% BSA in MTSB-T in concentration: 1:500 for primary and 1:1000 for secondary antibody. Between the all described steps the seedlings were washed 5×15 min with MTSB-T. For the final step, MTSB-T was exchanged by deionized water. From the robot seedlings were transferred to microscopy slides into 50% glycerol in deionized water and fluorescence signal was imaged by Zeiss LSM 880 inverted confocal scanning microscope equipped with Airyscan detector with 40×/1.2 C-Apochromat objective.

## Mesh penetration test

Four-day-old seedlings were transferred on ½ MS, 1% (wt/vol) plant agar, 1% (wt/vol) sucrose, pH 5.8, covered with sterilized nylon meshes of pore sizes of 109, 119, 139 µm (polyamide mesh UHELON). Plates were grown overnight vertically and then were tilted to 45° and grown 24 hr. After that, the root penetration through the mesh was scored using a binocular microscope. The penetration efficiency was calculated by normalizing each variant (genotype × pore size) to the Col-0 efficiency in the 139 µm variant in the particular repetition. In addition, the reference Col-0 was normalized to the Col-0 average over all repeats.

## Image analysis

We determined root elongation by measuring the total root length increment between the first and last time frame divided by the number of minutes. The root length increment was measured with the segmented line in ImageJ/Fiji v1.53f51 (*Schindelin et al., 2012*). The pH along the root was automatically measured from 10 to 25 pixels off the root surface using custom Python scripts (ATR v5, https://sourceforge.net/projects/atr-along-the-root, see *Figure 1—figure supplement 1f*) averaging bins of 15*20 pixels from the fluorescence intensity of both 405 and 488 nM channels. Root angles over time were automatically measured using ACORBA v1.2 (*Serre and Fendrych, 2022*). Fluorescence intensity of AFB1-mCitrine and CNGC14-GFP was measured as the signal intensity along a segmented line that covered the epidermal cell file using Fiji; the intensities were then binned to 10 µm bins and plotted.

## Statistical analysis and graphics

The averages and SD of surface pH or alkalinization factor in function of the distance from the root tip were plotted using the Sourcecode1.py Python script. The root elongation boxplots and the associated statistics were performed using the Sourcecode2.R: R script.

Statistical analyses were performed using the R software (R v4.0.2 and RStudio v1.3.1073). Boxplots represent the median and the first and third quartiles, and the whiskers extend to data points <1.5 interquartile range away from the first or third quartile; all data points are shown as individual dots. The results of the statistical tests are compiled in Supplemental statistics.

The pH surface profiles and data represented as boxplots were statistically analyzed from genotype to genotype or condition to condition using the nparcomp package (*Konietschke et al., 2015*). For two conditions or two genotypes comparisons, we used a non-parametric Student's test. For multiple comparisons, we used a non-parametric multi comparison test. For the pH surface profiles, each independent data point (a bin of pixels at a defined distance from the root tip) for the control genotype or condition were compared the same data point from the other(s) genotype(s) or condition(s) using the Sourcecode3.R or Sourcecode4.R R scripts. The results of the statistical tests for all the figures are compiled in *Source data 1*.

Sample size and biological replicates were determined as the maximum number of roots fitting the microscopy chamber, this number being influenced by the number of conditions and genotypes for a given experiment. Experiment technical replicates were arbitrarily set to three replicates. Fluorescence values were measured blindly by fully automated scripts. Data depicted in this publication represent a pool of technical and biological replicates. No technical or biological replicates were excluded from this publication.

Graphics were created with the R software or in Python (v3.8) with the Seaborn plugin (v0.11.2, https://doi.org/10.21105/joss.03021) and aesthetic modifications of the graphs (fonts, size) were modified in Inkscape (v1.0).

## Acknowledgements

We would like to thank Mark Estelle for sharing the seeds; Karel Müller and Tomáš Moravec for help with molecular cloning; Mayuri Sadoine and Wolf Frommer for access to their spectrofluorometer and technical guidance. Funding: This work was supported by the European Research Council (grant no. 803048) to MF and by the German Research Foundation (DFG Heisenberg Professorship; grant no. GR4559/4-1 and CRC1208 project A14) and Germany's Excellence Strategy (CEPLAS - EXC-2048/1 - project ID 390686111) to GG. AJ acknowledges the support of IEB imaging facility by MEYS CR LM2023050.

## Additional information

### Funding

| Funder | Grant reference number | Author |
|---|---|---|
| European Research Council | 803048 | Matyáš Fendrych |
| Deutsche Forschungsgemeinschaft | GR4559/4-1 | Guido Grossmann |
| Deutsche Forschungsgemeinschaft - CRC1208 | 267205415 | Guido Grossmann |
| CEPLAS-EXC-2048/1 | 390686111 | Guido Grossmann |

The funders had no role in study design, data collection and interpretation, or the decision to submit the work for publication.

### Author contributions

Nelson BC Serre, Conceptualization, Data curation, Software, Investigation, Visualization, Methodology, Writing – original draft, Writing – review and editing; Daša Wernerová, Conceptualization, Investigation, Methodology, Writing – original draft, Writing – review and editing; Pruthvi Vittal, Eva Medvecká, Adriana Jelínková, Investigation; Shiv Mani Dubey, Investigation, Methodology; Jan Petrášek, Guido Grossmann, Writing – review and editing; Matyáš Fendrych, Conceptualization, Supervision, Funding acquisition, Methodology, Writing – original draft, Writing – review and editing

### Author ORCIDs

Guido Grossmann ⬥ http://orcid.org/0000-0001-7529-9244
Matyáš Fendrych ⬥ http://orcid.org/0000-0002-9767-8699

### Decision letter and Author response

Decision letter https://doi.org/10.7554/eLife.85193.sa1

## Additional files

### Supplementary files

• MDAR checklist

• Source code 1. Python script used for plotting the surface pH as the funciton of distance from the root tip.

• Source code 2. R script used for generating the root elongation boxplots and the associated statistics.

• Source code 3. R script for comparing root surface pH profiles.

• Source code 4. R script for comparing root surface pH profiles.

• Source data 1. The results of the statistical tests for all the figures.

### Data availability

All source data used for the figure construction are provided as a supplementary files for each figure. The Python and R scripts are provided as Source codes. All raw data are available at Zenodo (main figures: https://doi.org/10.5281/zenodo.8138861; figure supplements: https://doi.org/10.5281/zenodo.8140893). Custom programs used in this article are available on their respective online repositories: ACORBA (https://sourceforge.net/projects/acorba/), ATR (https://sourceforge.net/projects/atr-along-the-root).

The following datasets were generated:

| Author(s) | Year | Dataset title | Dataset URL | Database and Identifier |
|---|---|---|---|---|
| Serre NBC, Wernerová D, Vittal P, Dubey SM, Medvecká E, Jelínková A, Petrášek J, Grossmann G, Fendrych M | 2023 | The AUX1-AFB1-CNGC14 module establishes a longitudinal root surface pH profile main figures | https://doi.org/10.5281/zenodo.8138861 | Zenodo, 10.5281/zenodo.8138861 |
| Serre NBC, Wernerová D, Vittal P, Dubey SM, Medvecká E, Jelínková A, Petrášek J, Grossmann G, Fendrych M | 2023 | The AUX1-AFB1-CNGC14 module establishes a longitudinal root surface pH profile supplementary figures | https://doi.org/10.5281/zenodo.8140893 | Zenodo, 10.5281/zenodo.8140893 |

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
