## [Editor Report]

All the results present solid evidence supporting the impact statement that 'Plant roots can rapidly change the acidity of their cell walls and the root-soil interface to efficiently navigate in the growing environment." These findings are important and have practical implications beyond Arabidopsis biology with potential future impacts in crop improvement, soil sciences and general plant physiology. The evidence is convincing and appropriately validated in line with current state-of-the-art.

---

## [Decision Letter]

**Decision letter after peer review:**

Thank you for submitting your article "The AUX1-AFB1-CNGC14 module establishes longitudinal root surface pH profile" for consideration by *eLife*. Your article has been reviewed by 3 peer reviewers, and the evaluation has been overseen by a Reviewing Editor and Jürgen Kleine-Vehn as the Senior Editor. The following individuals involved in the review of your submission have agreed to reveal their identity: Herman Höfte (Reviewer #1); Elke Barbez (Reviewer #3).

Essential revisions:

1) Please answer point by point the reviewers' comments below, and make the necessary changes in the manuscript. Some experiments are suggested to strengthen the results. If these can not be carried out, we request justification why.

2) A critical point highlighted by reviewers 1 and 2 is the Immunolabeling result with anti-AHA2. The experiment lacks good negative and positive controls (see reviewer 1) and quantification of the signal. Also, reviewer 2 indicates that the nature and specificity of the antibody have not been properly described.

*Reviewer #1 (Recommendations for the authors):*

A few remarks:

Figure 2A and S2A: the immunolabeling with the anti-AHA2 antibody is not convincing since good negative (aha mutants?) and positive controls (AHA overexpressor?) are lacking. Also, some quantification of the signal in WT and mutants should be provided. line 341: Please provide evidence for the dependence of FS fluorescence on other ions.

line 301ff: Would be nice to give an idea of the time scale of the pH variations in vertically growing roots? It appears that the fluorescence intensity variation on each side of the root also may show some periodicity.

*Reviewer #2 (Recommendations for the authors):*

– Improve the presentation of the more acidic and alkaline domains. In Figures 1e, and f it would be beneficial to highlight the three regions. In connection with the following point, measurements could be binned within these three areas and serve as a base for comparing treatments or genotypes.

– Visualisation of pH profile and statistical analysis. The profiles presented are great, but I am puzzled by how the quantitative comparisons between treatments/genotypes were computed and which statistical methods were used. Colored bars extend above regions of the profile and are marked as 'ns'. Does that mean that the profiles do not differ in these regions only but do in the other? I assume the color refers to the genotype/treatment that does not differ from the control. In any case, I wonder whether, in addition (or in replacement) to these profiles, the authors could not bin the measurement in three regions (apical, TZ, elongation zone) and present box plots of the ratio measurement.

– There are few details about the nature of the AHA antibody used to detect the AHA. Was it used in previous studies? Is it specific to the AHAs?

– Figure 4 pH profile in tir triple, cngc14, and afb1: move S4B, D, E and F to main figure.

– Figure 4G: enrichment of AFB1 and CNGC14 signal should be quantified.

*Reviewer #3 (Recommendations for the authors):*

Maybe the authors can still give it a try to provide a few complementary experiments with previously published tools. (Gjetting et al., doi: 10.1093/jxb/ers040., Barbez et al., as cited). Especially since Barbez et al., report a slight apoplast acidification in the transition zone shortly before cell elongation, it would be very interesting to read the authors' view on that

A suggestion regarding HPTS.

When a suitable 458 or white light laser is absent, one could try to calibrate the HPTS detection only with the 405 lasers. The pH unit resolution is lower but it may be appropriate for the authors' needs. In addition, HPTS provides, upon staining and mounting, a very strong background signal. Zeiss confocal setups often have signal overload detector protection which does not allow to image the stained roots as such. Therewith, it helps to detect the root in the DIC channel and crop the ROI in a way that no background (free medium) signal is visible in the channel. Therewith, one can increase the gain without risking damaging the detectors.

My favorite finding of this manuscript is the role of CNGC14 in the establishment of the alkalic root surface patch. It would be very interesting to see whether the authors can establish the patch elsewhere in the root by mis-expressing CNGC14 in the cngc14 background.

---

## [Author Response]

Essential revisions:Reviewer #1 (Recommendations for the authors):A few remarks:Figure 2A and S2A: the immunolabeling with the anti-AHA2 antibody is not convincing since good negative (aha mutants?) and positive controls (AHA overexpressor?) are lacking. Also, some quantification of the signal in WT and mutants should be provided.

We have modified the text to specify which AHA paralogs the antiAHA antibody recognizes (it recognizes multiple AHAs from multiple plant species), and we have performed additional controls: (A) omitting the primary antibody results in no PM staining. (B) in the aha2-4 mutant that lacks the AHA2 paralog, we can still detect plasma membrane signal. (C) in the inducible overexpression lines, the signal colocalizes with the AHA2-d95-mScarlet signal. These results confirm the antibody manufacturer specification – the antibody recognizes multiple AHA paralogs in Arabidopsis. We have included these results as Supplemental figures Figure S2-1.

We prefer not to compare the signal intensities between different immunostained roots, as the fluorescence intensity levels are quite variable. Instead, we included several examples of the immunostained mutants in the new Figure S2-1d for comparison. We also ‘softened’ the statements in the text where we refer to the comparison of immunostaining pattern of the aux1, afb1 and cngc14 mutants to the Col-0 control.

line 341: Please provide evidence for the dependence of FS fluorescence on other ions.

This data is shown as Figure S1e.

line 301ff: Would be nice to give an idea of the time scale of the pH variations in vertically growing roots? It appears that the fluorescence intensity variation on each side of the root also may show some periodicity.

The surface pH on the two sides of the root growing downward seem to be correlated – as is shown on the figure 5e. The pH seems to oscillate on the timescale of minutes, as is shown in Figure 5e and the Video 2. We have tried to characterize the oscillations using several methods (such as Fourier transformation of the signal), but we were unable obtain convincing results, the reason is that the oscillations are very variable and very likely depend on the growth patterns of the roots – we think that the roots is constantly correcting the growth directionality. We have added additional movies of vertically growing roots, so the readers can observe the dynamics of the surface pH patterns.

Reviewer #2 (Recommendations for the authors):– Improve the presentation of the more acidic and alkaline domains. In Figures 1e, and f it would be beneficial to highlight the three regions. In connection with the following point, measurements could be binned within these three areas and serve as a base for comparing treatments or genotypes.

We have included the domain highlight into the Figure 1. We have tried to explain the statistic comparison more clearly in the new version of the manuscript, and we have modified the figures to make to comparison clearer. We have put a lot of effort to measure and plot the longitudinal pH profiles of the roots (including a dedicated program to analyze the pH profiles). We are convinced that showing the full pH profiles is the best way how to show the results. The binning, albeit simpler to compare between treatments and genotypes, would have to be arbitrary and would hide a lot of information. We hope that the new explanation and marking the statistical significance improved the readability of our results.

– Visualisation of pH profile and statistical analysis. The profiles presented are great, but I am puzzled by how the quantitative comparisons between treatments/genotypes were computed and which statistical methods were used. Colored bars extend above regions of the profile and are marked as 'ns'. Does that mean that the profiles do not differ in these regions only but do in the other? I assume the color refers to the genotype/treatment that does not differ from the control. In any case, I wonder whether, in addition (or in replacement) to these profiles, the authors could not bin the measurement in three regions (apical, TZ, elongation zone) and present box plots of the ratio measurement.

We admit that we have not explained the statistical comparison well in the previous version of the manuscript, and that the visualization was quite confusing. We added clarifications in the text (figure legends and the material and methods) regarding how the statistics were done for profile comparisons: In the graphics, we quantified bins of fluorescence over the distance from the root tip. As these datapoints are independent (in contrary to e.g., timepoints) we used non-parametric Student test (2 conditions or genotypes) or multi comparison test (>2 conditions or genotypes) (nparcomp package for R). We compared every datapoints corresponding to all the biological/technical replicates for the control condition or genotype to the datapoints from another condition or genotype. To make the plots more readable, we modified all the pH profile plots in the manuscript and added graphical representation of the statistical results directly above the curves.

– There are few details about the nature of the AHA antibody used to detect the AHA. Was it used in previous studies? Is it specific to the AHAs?

We have added more information about the antiAHA antibody into the text and we performed additional control experiments that confirm what the producer claims: this antibody recognizes multiple AHA paralogs, including AHA2. These results are now part of the Figure S2-1.

– Figure 4 pH profile in tir triple, cngc14, and afb1: move S4B, D, E and F to main figure.– Figure 4G: enrichment of AFB1 and CNGC14 signal should be quantified.

We have reorganized the Figure 4 for better readability; the pH profiles of afb1, cngc14 and tir triple mutants are part of the main figure. In the supplemental figure, the comparison between the mutant profile and Col-0 is shown, while the main figure depicts the responses of the profile to application of IAA (reflecting the auxin responsiveness of the surface pH).

We have quantified the AFB1 and CNGC14 intensities along the longitudinal axis of the root, the data is now included as Figure 4h.

Reviewer #3 (Recommendations for the authors):Maybe the authors can still give it a try to provide a few complementary experiments with previously published tools. (Gjetting et al., doi: 10.1093/jxb/ers040., Barbez et al., as cited). Especially since Barbez et al., report a slight apoplast acidification in the transition zone shortly before cell elongation, it would be very interesting to read the authors' view on thatA suggestion regarding HPTS.When a suitable 458 or white light laser is absent, one could try to calibrate the HPTS detection only with the 405 lasers. The pH unit resolution is lower but it may be appropriate for the authors' needs. In addition, HPTS provides, upon staining and mounting, a very strong background signal. Zeiss confocal setups often have signal overload detector protection which does not allow to image the stained roots as such. Therewith, it helps to detect the root in the DIC channel and crop the ROI in a way that no background (free medium) signal is visible in the channel. Therewith, one can increase the gain without risking damaging the detectors.

We thank the reviewer for the suggestions regarding the other pH imaging tools. We have included the HPTS image into the supplementary Figure 1 – excited with 405 and 488 lasers. It is obvious that HPTS also demonstrates a similar pattern as FS: HPTS visualizes the strong alkaline domain around the transition zone (Figure S1a). However, the problem is that our vertical imaging system is a spinning disk microscope, and it is impossible to detect the cell wall signal due to the extremely strong HPTS signal around the root, which overshines the cell wall signal. Unlike the classical confocal, the spinning disk system illuminates the entire sample and regions of interest of illumination cannot be performed. We would therefore have to perform the imaging on a different vertical laser scanning confocal microscope that we do not have.

We are intensively working with other apoplastic pH staining methods and sensors (apophluorin, apohusion, SYP122-pHusion, acidins) and we are developing further rootoptimized pH sensors. So far, none of the published sensors was satisfactory in our hands. We, however, would prefer not to include this ‘negative’ data in this manuscript, as it seems beyond the scope of this work. We are aware that determining the correlation between cell wall pH, apoplastic pH and cell expansion is an important un-answered question. We have modified the discussion to include the comment of the reviewer:

“…, it remains to be resolved to what extent the surface pH correlates with the pH of the cell walls and the apoplastic space of the root epidermal cells, as previous work has reported a correlation between cell length and cell wall pH”.

My favorite finding of this manuscript is the role of CNGC14 in the establishment of the alkalic root surface patch. It would be very interesting to see whether the authors can establish the patch elsewhere in the root by mis-expressing CNGC14 in the cngc14 background.

This is an intriguing idea – we have tried to overexpress CNCG14 using a strong constitutive 35S promoter, but unfortunately, we were unable to obtain lines with strong CNGC14 signal (probably due to technical/cloning issues), we therefore do not have the results that could answer this question. We would need to target the expression into other domains of the root, and probably we would have to co-express AFB1 in this region as well.